# Optimal Parameter and Neuron Pruning for Out-of-Distribution Detection

**Chao Chen**[1], **Zhihang Fu**[1]*, **Kai Liu**[2], **Ze Chen**[1], **Mingyuan Tao**[1], **Jieping Ye**[1]

[1]Alibaba Cloud    [2]Zhejiang University

{ercong.cc, zhihang.fzh, yejieping}@alibaba-inc.com

## Abstract

For a machine learning model deployed in real world scenarios, the ability of detecting out-of-distribution (OOD) samples is indispensable and challenging. Most existing OOD detection methods focused on exploring advanced training skills or training-free tricks to prevent the model from yielding overconfident confidence score for unknown samples. The training-based methods require expensive training cost and rely on OOD samples which are not always available, while most training-free methods can not efficiently utilize the prior information from the training data. In this work, we propose an **O**ptimal **P**arameter and **N**euron **P**runing (**OPNP**) approach, which aims to identify and remove those parameters and neurons that lead to over-fitting. The main method is divided into two steps. In the first step, we evaluate the sensitivity of the model parameters and neurons by averaging gradients over all training samples. In the second step, the parameters and neurons with exceptionally large or close to zero sensitivities are removed for prediction. Our proposal is training-free, compatible with other post-hoc methods, and exploring the information from all training data. Extensive experiments are performed on multiple OOD detection tasks and model architectures, showing that our proposed OPNP consistently outperforms the existing methods by a large margin.

## 1 Introduction

Over the past decade, deep neural networks have achieved dramatic performance gains in computer vision [7], natural language processing [38], and AI for Science [25]. However, when deploying those deep learning models in real world scenarios, the model will provide a false prediction result for unseen categories, which will lead to serious security issues. A promising approach to address this problem is Out-of-Distribution (OOD) detection [51], which aims to distinguish whether the given sample is from training class or unknown category. The deep learning models with good OOD detection ability know what they don't know and can be used safely in real world applications.

Recently, a large number of works have been proposed to address the OOD detection problem [22, 21, 26]. Early representative methods utilized the Maximum Softmax Probability (MSP) [21] or Mahalanobis distance [26] as score function. The main challenge is that modern overparameterized deep neural networks can easily produce overconfident predictions on OOD samples, making the in-distribution (ID) data and OOD data inseparable. To alleviate the overconfident problem, some training-based methods and post-hoc methods have been proposed. The training-based methods mitigate the overconfident problem by incorporating OOD samples in training process [22] or synthesizing virtual outliers to regularize the model's decision boundary [9]. The post-hoc methods mainly focused on optimizing score function [24, 29, 31], or rectifying activations [43, 6] to make the ID and OOD samples more separable. The training-based methods are controllable and interpretable, but they rely on expensive training cost and additional OOD samples, which are not always available.

---

*Corresponding Author

37th Conference on Neural Information Processing Systems (NeurIPS 2023).

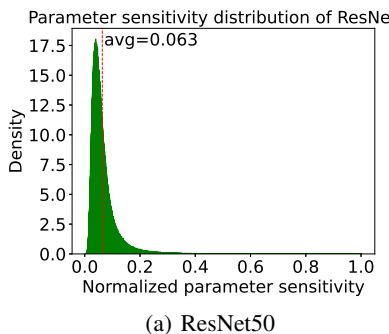
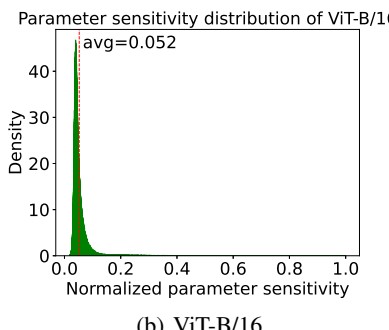

|  (a) ResNet50 | (b) ViT-B/16 |

Figure 1: Illustration of parameter sensitivity distribution for (a) ResNet50 and (b) ViT-B/16. The parameters are selected from the last fully-connected layer for both ResNet50 and ViT-B/16. The dotted line in red indicates the average sensitivity and the maximum sensitivity is normalized to 1.

In contrast, the post-hoc methods are training-free, low-cost and plug and play, however they can not effectively leverage the prior information from the training data and trained models.

It has been widely observed that overparameterized deep neural networks often suffer from redundant parameters and neurons, which consequently result in overconfident predictions. Conversely, a precisely pruned sub-network is capable of achieving comparable performance [13, 17, 5, 30, 46]. This motivates a straightforward question: *Can we identify the parameters and neurons that lead to overconfident outputs by leveraging the prior information from the off-the-shelf models and training samples ?* To answer this question, we first demonstrate the parameter sensitivity of the last fully-connected (FC) layer for ResNet50 [18] and Vision Transformer (ViT-B/16) [7] in Fig. 1. As can be observed, the distribution of parameter sensitivity is heavily positively skewed. More specifically, a significant proportion of parameter sensitivities are close to zero, while only a few parameters exhibit exceptionally high sensitivities. The gap between the maximum and minimum sensitivity values is more than 200 times.

The above observation naturally inspires a simple yet surprisingly effective method — **O**ptimal **P**arameter and **N**euron **P**runing (**OPNP**) for OOD detection. The motivations of our OPNP are in two aspects: (1) The parameters and neurons with sensitivities close to zero are redundant and can lead to overconfident predictions [23, 15, 44]. (2) The parameters with exceptionally large sensitivity result in a sharp landscape, which hurts the model generalization [11, 55, 54]. Therefore, in this study, we present empirical and theoretical evidences to show that the OOD detection performance can be significantly improved by pruning parameters and neurons with exceptionally large or close to zero sensitivities. The main contribution of this paper are as follows:

- A gradient based approach is present to estimate the sensitivity of parameters and neurons in deep model. Building upon this approach, we introduce OPNP - A simple yet effective training-free method, which significantly improves OOD detection performance by removing weights and neurons with exceptionally large or close to zero sensitivities.

- We evaluate OPNP on different OOD detection tasks and model architectures, including ResNet and ViT. Compared to the baseline model, OPNP achieves 32.5% FPR95 reduction on a large-scale ImageNet-1k benchmark, and outperforms existing state-of-the-art post-hoc OOD detection method by 5.5% in FPR95.

- Extensive ablation experiments are performed to reveal the insight and effectiveness of the proposed method. We show that OPNP is compatible with other post-hoc methods and can benefit model calibration. We believe our insights can inspire and accelerate future research in related tasks.

## 2 Related Work

**General OOD Detection.** OOD detection is highly relevant to several early research tasks, including outlier detection (OD), anomaly detection (AD) [40] and open-set recognition (OSR) [14]. All

these tasks are aim to identifying the OOD samples in the open world scenario [51]. The most promising OOD detection methods can be divided into five categories, including density-based methods [39], distance-based methods [26, 45], outlier exposure methods [22, 28, 2], virtual OOD synthesis methods [9, 47], and post-hoc methods [31, 43]. Besides, some recent advances also pay attention to exploring large-scale pretrained vision-language model for OOD detection [35, 10], and extending OOD detection from classification to other learning tasks, such as object detection [8] and segmentation [20]. In this study, we mainly focus on post-hoc OOD detection, which is a training-free approach that does not require any OOD samples. For a more comprehensive understanding of the general OOD detection task, we suggest referring to [51] for further details.

**Post-hoc OOD Detection.** Recently, the post-hoc OOD detection methods have achieved promising performance and drawn increasing attention [44, 43, 57, 31, 29]. To alleviate the over-confident prediction caused by the softmax function, ODIN [29] introduces a temperature scaling strategy to make the softmax scores between ID and OOD images more separable. Liu et al. [31] replace the MSP score with energy score, which is theoretically aligned with the probability density and less susceptible to the overconfident problem. In huang et. al [24], GradNorm is presented which utilizes the magnitude of gradients as OOD score. Another line of work relieve overconfident problem by rectifying typical features in the penultimate layer [43, 6, 57]. In particular, ReAct [43] truncates features with a global threshold, which is chosen to preserve the activations for ID data while rectifying that of OOD data. ASH [6] shows that simply removing lower activations or binarizing representations leads to promising OOD detection performance. The most relevant method to our proposal is DICE [44], which ranks weights based on a measure of contribution, and selectively use the most salient weights to derive the output for OOD detection. Both DICE and our proposal alleviate over-fitting by model pruning. DICE selects the most important connections while our proposal removes the most sensitive and insensitive parameters and neurons. Besides, the method used to measure the weight importance or sensitivity is different.

**Parameter and Neuron Pruning.** In deep neural networks, parameter and neuron pruning is widely used for reducing over-fitting [23, 42, 15, 5] and model compression [33, 17, 27] . Dropout is the most well-known method to improve the robustness of deep models, which randomly drops some neurons [42] or connections [48] in training time. Based on dropout, Gomez et al. [15] introduce targeted dropout, which drops units and weights with low magnitude. They experimentally demonstrated that target dropout benefit to post-hoc pruning of units and weights. Besides, parameter and neuron pruning have also been widely used in model compression. Han et al. [17] indicate that the weights with low magnitude is less important and propose to remove those weights for model compression. In contrast, Li et al. [27] propose to drop the feature maps based on weight norms, the model performance is well preserved after pruning more than 30% units. In [30, 3], the authors demonstrate the effectiveness of ensembling multiple sparse networks, which are trained from scratch with dynamic sparsity constraint, for OOD detection. The aforementioned researches have demonstrated that the weights and neurons are significantly redundant in deep networks, which inspires us to prune the parameters and neurons that result in over-fitting for OOD detection. We believe optimal post-hoc parameter and neuron pruning is a promising approach for OOD detection.

## 3 Method

### 3.1 Problem statement

Assume that we have an in-distribution dataset $\mathcal{D}_{in}$ of pairs $(\boldsymbol{x}_{in}, y_{in})$ and an out-of-distribution dataset $\mathcal{D}_{out}$ of pairs $(\boldsymbol{x}_{out}, y_{out})$, where $x_{in}, x_{out} \in \mathcal{X}$ denote the input feature vector of ID and OOD samples, $y_{in} \in \mathcal{Y}_{in} := \{1, 2, \cdots, K\}$ denotes the ID class label, and $y_{out} \in \mathcal{Y}_{out}$ denotes the output class label, $\mathcal{Y}_{in} \cap \mathcal{Y}_{out} = \emptyset$. Given a classification model $f(\boldsymbol{x}; \boldsymbol{\theta})$ trained from in-distribution dataset $\mathcal{D}_{in}$. The goal of post-hoc OOD detection is to design a binary classifier $g_\lambda(x)$ which is able to distinguish whether the test sample is from ID or OOD distribution. Therefore, the challenge of OOD detection is to find an optimal score function $S(\boldsymbol{x})$ such that for a given test sample $x \in \mathcal{X}$,

$$g_\lambda(x) = \begin{cases} ID, & S(\boldsymbol{x}) \geq \lambda \\ OOD, & S(\boldsymbol{x}) < \lambda \end{cases} \tag{1}$$

where samples with higher score $S(\boldsymbol{x})$ are classified as ID and vice versa, $\lambda$ is the threshold which usually set to ensure 95% ID samples are correctly classified. Maximum softmax probability [21]

and energy score [31] are the most widely used score functions in post-hoc OOD detection. We follow previous post-hoc methods [43, 44] to utilize energy score as OOD detection metric, which consistently outperforms MSP score.

We denote $h(\boldsymbol{x}) \in \mathbb{R}^L$ the feature representation from the penultimate layer, denote $\mathbf{W} \in \mathbb{R}^{L \times K}$ and $\mathbf{b} \in \mathbb{R}^K$ the output weights and bias of the last FC layer. Then, the output logit can be given as

$$f(\boldsymbol{x}; \boldsymbol{\theta}) = \mathbf{W}^\top h(\boldsymbol{x}) + \mathbf{b} \tag{2}$$

The energy score function [31] maps the output logit to a energy value by,

$$E(\boldsymbol{x}; \boldsymbol{\theta}) = -\log \sum_{i=1}^{K} \exp(f_i(\boldsymbol{x})) \tag{3}$$

where $f_i(\boldsymbol{x})$ denotes the logit output for class $i$. The energy score reduces over-fitting caused by softmax function, but the overparameterized connection weights $\mathbf{W}$ in the last fully-connected layer may still cause overconfident logit output. Therefore, in the following section, we aim to provide an optimal parameter and neuron pruning strategy to reduce over-fitting.

## 3.2 Parameter sensitivity estimation

In this section, we propose to estimate parameter sensitivity by measuring how sensitive the output energy score to a small change of parameters. For a given sample $\boldsymbol{x}_k$ and corresponding energy output $E(\boldsymbol{x}_k; \boldsymbol{\theta})$, a small change $\boldsymbol{\delta}_{ij}$ is added to the parameter $\boldsymbol{\theta}_{ij}$, which results in a change in the output energy score,

$$E(\boldsymbol{x}_k; \boldsymbol{\theta} + \boldsymbol{\delta}) - E(\boldsymbol{x}_k; \boldsymbol{\theta}) \approx \sum_{i,j} g_{ij}(\boldsymbol{x}_k) \boldsymbol{\delta}_{ij} \tag{4}$$

$$g_{ij}(\boldsymbol{x}_k) = \frac{\partial E(\boldsymbol{x}_k; \boldsymbol{\theta})}{\partial \boldsymbol{\theta}_{ij}} \tag{5}$$

Here, $g_{ij}(\boldsymbol{x}_k)$ denotes the gradient of model output to the parameter $\boldsymbol{\theta}_{ij}$ at data point $\boldsymbol{x}_k$. Since $\boldsymbol{\delta}_{ij}$ is a small constant, the parameter sensitivity to model output can be measured by the magnitude of the gradient $g_{ij}$. In this respect, given a batch of samples $\{\boldsymbol{x}_k\}_{k=1}^m$, the parameter sensitivity can be estimated by accumulating the gradients over all input samples,

$$\mathbf{M}_{ij} = \frac{1}{m} \sum_{k=1}^{m} |g_{ij}(\boldsymbol{x}_k)| \tag{6}$$

where $\mathbf{M}_{ij}$ denotes the sensitivity of parameter $\boldsymbol{\theta}_{ij}$. It's worth noting that we utilize the change in energy score as the sensitivity measure, which is better aligned with the OOD detection metric. Practically, the parameter sensitivity can also be measured by other model output, such as the change of logit norm $\|f(\boldsymbol{x}; \boldsymbol{\theta})\|_2$, which has been exploited in lifelong learning [1].

In Fig. 1, we illustrate the sensitivity of parameter $\mathbf{W}$ for two representative deep networks ResNet50 [18] and ViT-B/16 [7], where the maximum sensitivity is normalized to 1. It can be seen that the maximum parameter sensitivity of the ResNet50 and ViT-B/16 is nearly 20 times than the average sensitivity. In addition, compared with ViT-B/16, there are more parameters with sensitivity close to zero in ResNet50, which indicates that the last FC layer of ResNet50 has more redundant parameters. Based on the intuition that the parameters and neurons with exceptionally large sensitivity or with sensitivity close to zero tend to result in overconfident prediction, an optimal parameter and neuron pruning strategy is introduced to improve the OOD detection performance.

## 3.3 Optimal parameter and neuron pruning

In this section, we mainly introduce how to prune the connection weights and neurons in the last fully-connected layer, which directly result in overconfident results. The pruning strategy in other layers can be achieved in the same manner.

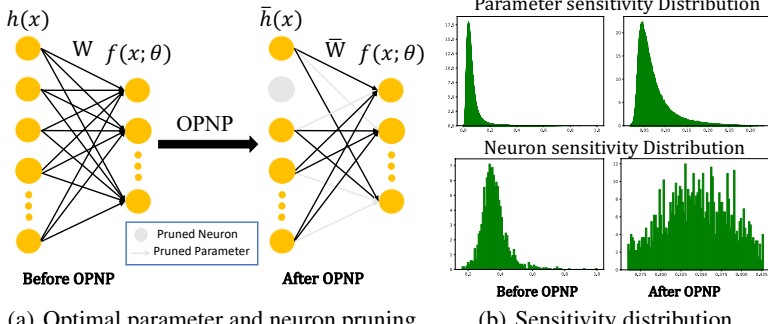

(a) Optimal parameter and neuron pruning     (b) Sensitivity distribution

Figure 2: (a) Illustration of the last fully-connected layer before and after OPNP, the connections and neurons in grey color represent the pruned ones. (b) The first row illustrate the parameter sensitivity before and after pruning and the second row illustrate the neuron sensitivity before and after pruning.

**Parameter Pruning.** Given the connection weights $\mathbf{W}$ that maps the feature representation to logit, the corresponding parameter sensitivity $\mathbf{M}$ can be computed according to Eq. 6. As mentioned above, the parameters with exceptionally large or close to zero sensitivity tend to result in over-fitting. Therefore, a simple threshold function can be utilized to remove the risky connections,

$$\overline{\mathbf{W}}_{ij} = \begin{cases} 0, & \mathbf{M}_{ij} < \Omega^w_{min} \\ 0, & \mathbf{M}_{ij} > \Omega^w_{max} \\ \mathbf{W}_{ij} & \text{other} \end{cases} \tag{7}$$

where $\overline{\mathbf{W}}_{ij}$ denotes the weights after pruning, $\Omega^w_{min}$ and $\Omega^w_{max}$ denote the minimum and maximum thresholds. To align with previous post-hoc methods [43, 44, 6], we obtain the threshold by a percentile $\rho$, which indicates that the threshold is set to the $\rho$th-percentile of the entire sensitivity matrix. For example, $\rho^w_{max} = 1\%$ represents that $\Omega^w_{max}$ is set to the 1% largest sensitivity value in $\mathbf{M}$, and $\rho^w_{min} = 10\%$ represents that $\Omega^w_{min}$ is set to the 10% smallest sensitivity value in $\mathbf{M}$. In Fig. 2(b), the first row illustrates the sensitivity distribution of parameter $\mathbf{W}$ in ResNet50 before and after parameter pruning. After pruning, the connection weights that larger than $\Omega^w_{max}$ or smaller than $\Omega^w_{min}$ are removed, and will not contribute to the output logit, which reduces the risk of over-fitting.

**Neuron Pruning.** In additional to parameter pruning, we also propose to prune the neurons in the pre-logit layer, which has also been shown to mitigate over-fitting [42, 15]. For the $i$-th neuron in the pre-logit layer, it contributes to all output neurons, therefore the sensitivity of the $i$-th neuron should be defined based on the sensitivity of the connection weights between the $i$-th hidden neuron and all output neurons. Considering that the $\ell_1$ or $\ell_2$ norm of the weights are usually used to measure the importance of the units in deep networks [41, 15]. We define the neuron sensitivity as the average sensitivity of weights that connected with the neuron, which is equivalent to $\ell_1$ norm of the weight sensitivity, and reflects the average sensitivity to all classes, i.e.,

$$\mathbf{O}_i = \frac{1}{K} \sum_{p=1}^{K} \mathbf{M}_{ip} \tag{8}$$

where $\mathbf{O}_i$ denotes the sensitivity of $i$-th neuron in pre-logit layer, $K$ represents the number of output neurons. In this respect, we can use a similar threshold function as Eq. 7 to remove the risky neurons,

$$\overline{h}^i(\boldsymbol{x}) = \begin{cases} 0, & \mathbf{O}_i < \Omega^o_{min} \\ 0, & \mathbf{O}_i > \Omega^o_{max} \\ h^i(\boldsymbol{x}) & \text{other} \end{cases} \tag{9}$$

where $\overline{h}^i(\boldsymbol{x})$ denotes the output feature from the $i$-th pruned neuron in the pre-logit layer, $\Omega^o_{min}$ and $\Omega^o_{max}$ represent the minimum and maximum sensitivity thresholds determined by the pruning percentage $\rho^o_{min}$ and $\rho^o_{max}$. The second row in Fig. 2(b) illustrates the sensitivity distribution of the hidden neurons in ResNet50 before and after neuron pruning, where the maximum sensitivity is

normalized to 1. As observed, after neuron pruning, the redundant neurons (with sensitivity close to zero) and risky neurons (with sensitivity far above the average) are removed and the distribution of sensitivity across neurons becomes more uniform, which potentially reduces over-fitting.

## 3.4 Insight Justification

The following remarks are provided to explain why OPNP improves OOD detection performance.

**Remark 1. Parameter and neuron pruning avoid overconfident predictions.** The over-parameterized deep neural networks tend to generate overconfident predictions even for OOD samples [19, 37, 16]. Therefore, most existing methods improve OOD performance by avoiding overconfident predictions [29, 31, 44]. For a deep network, the last fully connected layer can be regarded as a linear classifier. The most widely used technique to prevent a classifier from overfitting is to employ a $\ell_1$ or $\ell_2$ regularization, which can be formulated as $\min_{\boldsymbol{\theta}} \mathbb{E}_{(\boldsymbol{x},\boldsymbol{y}) \sim D} \|\boldsymbol{\theta}^\top \cdot h(\boldsymbol{x}) - \boldsymbol{y}\|_2^2 + \lambda \mathcal{R}(\boldsymbol{\theta})$, where $\mathcal{R}(\boldsymbol{\theta})$ represents $\ell_1$- or $\ell_2$-norm of $\boldsymbol{\theta}$. As the parameter pruning is able to reduce $\mathcal{R}(\boldsymbol{\theta})$, it can be regarded as an effective post regularization technique that reduces the model complexity and avoids overconfident predictions. Besides, the neuron pruning is similar to target dropout [15] which has also been demonstrated to reduce overfitting. Therefore, the proposed OPNP avoids overconfident predictions and potentially improves the OOD detection performance.

**Remark 2. Pruning the least sensitive parameters and neurons improve separability between ID and OOD samples.** We denote $f_j(x)$ the logit output of $j$-th class, after pruning the least sensitive parameters, the logit reduction of the $j$-th class can be estimated as

$$\Delta f_j(\boldsymbol{x}) = \sum_{\mathbf{M}_{jk} < \Omega_{min}^w} \mathbf{M}_{jk} \cdot |\mathbf{W}_{jk}| \cdot h_k(\boldsymbol{x}) \tag{10}$$

It shows that the logit reduction is positively correlated with the average sensitivity of the pruned weights. As the parameter sensitivity is computed over the training ID set, the least sensitive parameters on ID distribution should be more sensitive for OOD samples on average, i.e.,

$$\sum_{\mathbf{M}_{jk} < \Omega_{min}^w} \mathbf{M}_{jk}^{OOD} > \sum_{\mathbf{M}_{jk} < \Omega_{min}^w} \mathbf{M}_{jk}^{ID} \tag{11}$$

Therefore, the logit reduction on OOD samples is larger than on ID samples $\Delta f_j(\boldsymbol{x}_{out}) > \Delta f_j(\boldsymbol{x}_{in})$, which leads to better separability between ID and OOD samples, and improves OOD detection performance. We also show the parameter sensitivity distribution (on ID and OOD sets) of the pruned weights in Fig. 9, which experimentally verifies Eq. 11.

**Remark 3. Pruning the most sensitive parameters and neurons improves generalization.** We follow [54] to define the first-order flatness as

$$R_\rho(\boldsymbol{\theta}) \triangleq \rho \cdot \max_{\boldsymbol{\theta}' \in B(\boldsymbol{\theta}, \rho)} \|\Delta f(\boldsymbol{\theta}')\|, \quad \forall \boldsymbol{\theta} \in \boldsymbol{\Theta} \tag{12}$$

where $\Delta f(\boldsymbol{\theta}')$ denotes the derivative at point $\boldsymbol{\theta}'$, $B(\boldsymbol{\theta}, \rho) = \{\boldsymbol{\theta}' : \|\boldsymbol{\theta} - \boldsymbol{\theta}'\| < \rho\}$ denotes the open ball of radius $\rho$ centered at the point $\boldsymbol{\theta}$ in the Euclidean space and $\rho$ denotes the perturbation radius that controls the magnitude of the neighbourhood. The flatness $R_\rho(\boldsymbol{\theta})$ describes how flat the function landscape is [54]. It has been demonstrated that a flatter landscape could lead to better generalization [11, 55, 54, 53]. Eq. 12 indicates that the first-order flatness is determined by the largest gradient, therefore, our proposed method pruning the most sensitive parameters is able to improve the flatness of the function landscape and lead to better generalization. However, according to **Remark 2**, pruning the most sensitive parameters and neurons may also hurt the separability between ID and OOD samples. Therefore, there is a trade-off between better generalization and better ID-OOD separability. This explains why OOD performance improves with very few sensitive parameters pruned and drops with a large pruning ratio, as demonstrated in Fig. 3.

## 4 Experiments

In this section, we describe our experimental setup and implementation details, then evaluate the effectiveness of the proposed OPNP method in different model architectures and OOD detection benchmarks, followed by extensive ablation studies.

Table 1: OOD detection results on ImageNet-1k benchmark with ResNet50 model. OPP, ONP and OPNP represent only using optimal parameter pruning, only using optimal neuron pruning and using both parameter and neuron pruning, respectively. All numbers are percentages.

| Method | OOD Datasets | | | | | | | | Average | |
| | iNatualist | | SUN | | Places | | Texture | | | |
| | FPR95↓ | AUROC↑ | FPR95↓ | AUROC↑ | FPR95↓ | AUROC↑ | FPR95↓ | AUROC↑ | FPR95↓ | AUROC↑ |
|---|---|---|---|---|---|---|---|---|---|---|
| MSP[12] | 54.05 | 87.43 | 73.37 | 78.03 | 72.98 | 78.03 | 68.85 | 79.06 | 67.31 | 80.64 |
| ODIN[29] | 47.66 | 89.66 | 60.15 | 84.59 | 67.89 | 81.78 | 50.23 | 85.62 | 56.48 | 85.41 |
| Mahalanobis[26] | 97.00 | 52.65 | 98.50 | 42.41 | 98.40 | 41.79 | 55.80 | 85.01 | 87.43 | 55.47 |
| Energy[31] | 55.72 | 89.95 | 59.26 | 85.89 | 64.92 | 82.86 | 53.72 | 85.99 | 58.41 | 86.17 |
| ReAct[43] | 20.38 | 96.22 | 24.20 | 94.20 | 33.85 | 91.58 | 47.30 | 89.80 | 31.43 | 92.95 |
| DICE[44] | 25.63 | 94.49 | 35.15 | 90.83 | 46.49 | 87.48 | 31.72 | 90.30 | 34.75 | 90.77 |
| DICE+ReAct[44] | 18.64 | 96.24 | 25.45 | 93.94 | 36.86 | 90.67 | 28.07 | 92.74 | 27.25 | 93.40 |
| **OPP** | 23.58 | 95.41 | 30.40 | 93.17 | 40.76 | 90.65 | 41.27 | 92.10 | 34.00 | 92.83 |
| **ONP** | 18.56 | 95.93 | 26.67 | 94.72 | 32.69 | 92.94 | 38.56 | 90.83 | 29.12 | 93.61 |
| **OPNP** | 18.89 | 96.03 | **18.50** | 95.62 | **30.14** | **93.46** | 36.17 | 91.70 | 25.93 | 94.20 |
| **OPNP+ReAct** | 14.72 | **96.78** | 19.73 | **95.65** | 30.23 | 93.34 | **27.78** | **94.13** | **23.12** | **94.98** |

## 4.1 Experimental Setup

**Datasets.** Following prior works [43, 44], we utilize ImageNet-1K, CIFAR-10 and CIFAR-100 as ID dataset. For ImageNet-1K benchmark, 50000 test samples are used as test ID samples, and four datasets are used as test OOD data, which are from (subset of) iNaturalist, SUN, Place and Texture [44]. For CIFAR benchmark, the standard split with 50,000 training ID images and 10,000 test ID images are utilized for training and evaluation. We follow [43, 44] to evaluate on six common OOD datasets [2], including iSUN [50], LSUN-Resize [52], LSUN-Crop [52], SVHN [36], Places365 [56], and Textures [4].

**Models.** We utilize two representative deep models, ResNet50 [18] and ViT-B/16 [7], to evaluate our proposal. For ResNet50, the model weights provided in TorchVision [34] is utilized. For ViT-B/16, we utilize the model weights provided in Pytorch Image Models (timm) library [49]. For both models, we only estimate the parameter and neuron sensitivity in the last fully-connected (FC) layer. The number of neurons in th pre-logit layer are 2048 and 768 for ResNet50 and ViT-B/16 respectively.

**Evaluation Metric and Baselines.** We utilize FPR95 and AUROC as evaluation metrics, which are the most important metrics in OOD detection [22, 31]. FPR95 is short for FPR@TPR95 which represents the false positive rate when the true positive rate is 95%. AUROC denotes the area under the receiver operating characteristic curve, which is threshold-free and reflects the average performance under different thresholds. Note that we following [44] do not report the ID classification accuracy since our proposal is training-free and only revises the last FC layer. We can always use the original FC layer for classification, which ensures an identical classification accuracy as unpruned model. We compared our OPNP with the most competitive post-hoc OOD detection methods, including MSP [21], ODIN[29], Mahalanobis distance [26], Energy Score [31], ReAct [43] and DICE [44].

**Implementation Details.** Implementation of this work is based on Pytorch library [3]. We use all training images in ImageNet-1K to obtain the parameter sensitivity by using the *autograd* function achieved by *energy_score.backward()*. At test time, all images are resized to $224 \times 224$. The hyperparameters $\rho^w_{min}$, $\rho^w_{max}$, $\rho^o_{min}$, $\rho^o_{max}$ are experimentally determined in the validation set, which includes 50000 test ID samples and 50000 test OOD samples that are selected from images-21k. We utilize grid search to determine the optimal pruning percentage, where we vary $\rho^w_{min} = \{0, 5, 10, 20, \cdots, 60\}$, $\rho^w_{max} = \{0, 0.1, 0.3, 0.5, 1, 3, 5\}$, $\rho^o_{min} = \{0, 5, 10, 20, \cdots, 50\}$, $\rho^o_{max} = \{0, 0.5, 1, 5, 10, 20, \cdots, 50\}$. The same hyperparameters are adopted in the same model.

## 4.2 Main Results.

**Evaluation on ImageNet-1K benchmark.** In Table 1, we compare our proposal with other competitive post-hoc OOD detection methods based on ResNet50. ↑ denotes larger values are better and ↓ denotes smaller values are better. The results for those comparison baselines are directly taken from

[2]https://github.com/deeplearning-wisc/dice

[3]https://github.com/pytorch/pytorch

Table 2: OOD detection results on ImageNet-1k benchmark with ViT-B/16 model. All numbers are percentages.

| Method | OOD Datasets | | | | | | | | Average | |
| | iNatualist | | SUN | | Places | | Texture | | | |
| | FPR95↓ | AUROC↑ | FPR95↓ | AUROC↑ | FPR95↓ | AUROC↑ | FPR95↓ | AUROC↑ | FPR95↓ | AUROC↑ |
|---|---|---|---|---|---|---|---|---|---|---|
| MSP[12] | 21.28 | 91.63 | 51.96 | 85.08 | 50.63 | 84.92 | 50.57 | 87.50 | 43.61 | 87.28 |
| Energy[31] | 7.61 | 98.23 | 40.30 | 90.77 | 46.89 | 88.62 | **33.54** | **93.21** | 32.09 | 92.71 |
| ReAct[43] | **2.28** | **99.42** | 30.68 | 93.94 | 35.32 | 91.40 | 37.08 | 92.84 | 26.34 | 94.40 |
| DICE[44] | 4.51 | 98.87 | 32.43 | 93.30 | 37.46 | 91.02 | 39.19 | 92.44 | 28.40 | 93.91 |
| DICE+ReAct[44] | 2.65 | 99.38 | 29.45 | 93.52 | 38.45 | 91.17 | 33.78 | 93.27 | 26.08 | 94.34 |
| **OPP** | 3.12 | 99.18 | 25.28 | 93.99 | 34.00 | 91.43 | 35.56 | 92.21 | 24.49 | 94.20 |
| **ONP** | 3.87 | 99.13 | 29.63 | 93.08 | 35.68 | 90.73 | 35.60 | 91.54 | 26.20 | 93.62 |
| **OPNP** | 3.16 | 99.38 | 24.32 | 93.86 | 34.52 | 91.45 | 38.76 | 91.96 | 25.19 | 94.16 |
| **OPP+ReAct** | 2.52 | 99.35 | **23.96** | **94.50** | **32.80** | **92.10** | 36.03 | 91.77 | **23.83** | **94.43** |

Table 3: Changes in ID classification accuracy by varying pruning percentage in ResNet50 model.

| Percentage | $\rho^w = 0$ | $\rho^w_{max} = 0.1$ | $\rho^w_{max} = 0.3$ | $\rho^w_{max} = 1$ | $\rho^w_{max} = 5$ | $\rho^w_{min} = 5$ | $\rho^w_{min} = 10$ | $\rho^w_{min} = 20$ | $\rho^w_{min} = 40$ |
|---|---|---|---|---|---|---|---|---|---|
| **ID Acc** | 76.13 | 75.14 | 74.72 | 71.60 | 56.37 | 76.06 | 75.88 | 75.86 | 75.06 |

[43, 44], which utilizes the same experimental setup as ours. OPP, ONP and OPNP represent only utilize optimal parameter pruning, only utilize optimal neuron pruning and utilize both parameter and neuron pruning, respectively. The results in Table 1 reveal several interesting observations: (1) Optimal parameter pruning (OPP) achieves similar performance as DICE which also removes unimportant connections for OOD detection, and significantly outperforms MSP [21], Mahalanobis distance [26] and Energy [31] baselines. (2) Optimal neuron pruning (OPN) achieves much better performance than OPP, outperforms DICE [44] by a large margin and outperforms SOTA feature rectification method (ReAct) [43] by 2.1% in FPR95 and 0.7% in AUROC. (3) Compared to OPP and ONP, combining parameter pruning and neuron pruning (OPNP) also reduces FPR95 by 3.3% and improves AUROC by 0.6% based on ONP. (4) OPNP outperforms both SOTA weights pruning method (DICE) and SOTA feature rectification method (ReAct) by a large margin (more than 5% in FPR95). (5) Our proposed OPNP does not outperform ReAct and DICE in Texture dataset, which can be compensated by combining ReAct.

In Table 2, we compare our proposal with competitive post-hoc OOD detection methods based on ViT-B/16 model. We note that the performance of MSP [21] and Energy Score [31] in previous work [9] is worse than our implementation based on the same model, therefore, we reported the performance reproduced by ourselves. The results show that: (1) Both OPP and ONP outperforms Energy baseline [31] by a large margin, which demonstrates the effectiveness of our proposal in ViT model. (2) OPP outperforms ONP and combining parameter and neuron pruning (OPNP) does not further improve the OOD performance, which is different from the results in ResNet50. We think this is because the number of neurons in ViT-B/16 (768) is much less than in ResNet50 (2048), besides, there is a relu layer before pre-logit layer in ResNet, which results in more risky parameters and neurons in ResNet50 model. (3) OPP outperforms ReAct by 1.85% and outperforms DICE by 3.91%in FPR95, combining OPP and ReAct brings additional improvement.

**Evaluation on CIFAR benchmark.** The main results on CIFAR10 and CIFAR100 benchmarks are show in Table 5 and Table 6. As can be seen, we consider the three most commonly used post-hoc methods and compare the OOD detection performance with and without applying the OPNP. The results show that: (1) On both CIFAR10 and CIFAR100, using OPNP consistently outperforms the counterpart without OPNP, which indicates that our proposal is compatible with other post-hoc methods. (2) The performance improvement brought by OPNP on CIFAR100 is more significant than on CIFAR10 and less significant than on ImageNet benchmark. We believe this is because the FC layer on CIFAR10 classification model is much less overparameterized than CIFAR100 and ImageNet classification models. (3) The OPNP achieves similar performance as ReAct On CIFAR10 benchmark, and outperforms ReAct on CIFAR100 benchmark. Besides, utilizing OPNP and ReAct jointly reduces FPR95 by 1.55% and 2.49% in CIFAR10 and CIFAR100 benchmark, respectively.

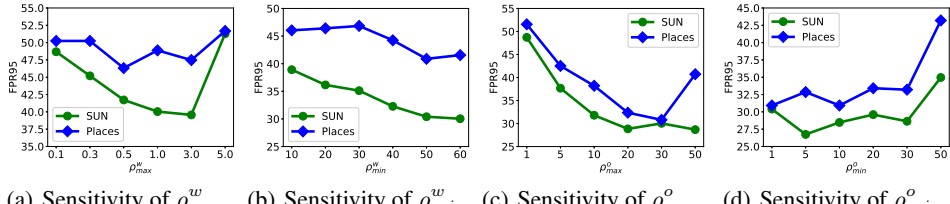

(a) Sensitivity of $\rho_{max}^w$    (b) Sensitivity of $\rho_{min}^w$    (c) Sensitivity of $\rho_{max}^o$    (d) Sensitivity of $\rho_{min}^o$

Figure 3: Effect of varying pruning percentage parameters in ResNet50 model. (a) Effect of varying $\rho_{max}^w$; (b) Effect of varying $\rho_{min}^w$ when set $\rho_{max}^w = 0.5$; (c) Effect of varying $\rho_{max}^o$; (d) Effect of varying $\rho_{min}^o$ when set $\rho_{max}^o = 30$. All numbers are percentages.

Table 4: OOD detection performance with different parameter and neuron pruning methods. We use ImageNet-1K as ID data, SUN and Places as OOD data. FPR95 performance in ResNet50 is reported.

| Method | RPP | TPP | OPP | RNP | TNP | ONP |
|--------|-------|-------|-------|-------|-------|---------|
| **SUN** | 49.42 | 43.36 | 30.40 | 52.44 | 47.76 | **26.67** |
| **Places** | 54.40 | 51.86 | 40.76 | 53.92 | 52.84 | **32.69** |

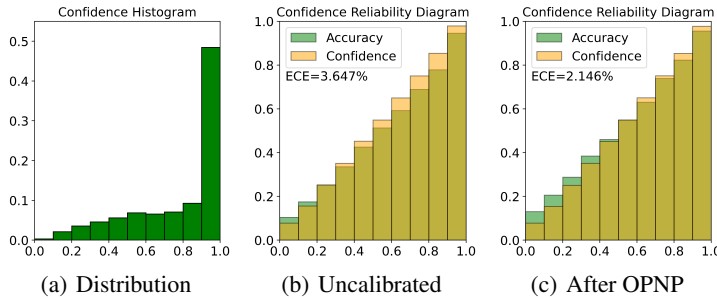

(a) Distribution      (b) Uncalibrated      (c) After OPNP

Figure 4: Illustration of confidence reliability diagrams. (a) Sample distribution histogram in different confidence bins. (b) Confidence reliability diagrams (CRD) in the original calibrated model. (c) CRD in the model with optimal parameter and neuron pruning.

### 4.3 Ablation Studies.

**Effect of pruning percentage.** In Fig. 3, we demonstrate the impact on OOD detection performance by varying pruning percentages in two different tasks. Fig. 3(a) shows that pruning only 0.5% high sensitivity parameters brings significant improvement. In Fig. 3(b), we observe considerable performance improvement with a large pruning percentage for low sensitive parameters. Fig. 3(c) and Fig. 3(d) suggest that pruning high sensitive neurons is more effective than pruning low sensitive neurons, which brings marginal improvement. While OPNP achieves the SOTA performance, the performance could be significantly improved by pruning only the weights or neurons, which is much simple to determine the thresholds. Besides, the performance is improved and insensitive in a wide range of pruning ratio. For example, the optimal pruning ratio can be set to $\rho_{min}^w \in [10, 30]$ and $\rho_{max}^w \in [0.5, 3]$ across different OOD sets. In Tabel 3, we demonstrate the impact of different pruning percentages on ID classification accuracy. As observed, pruning only 1.0% high sensitive parameters decreases the ID accuracy by 4.53%. In contrast, pruning 40% low sensitive parameters only reduces ID accuracy by 1.07%. This highlights the effectiveness of our sensitivity estimation method, and also demonstrates the significant parameter redundancy in deep networks.

**Ablation on pruning methods.** In this ablation, we compare the proposed sensitivity guided parameter and neuron pruning method with other pruning method, including: (1) Random parameter pruning (RPP) [48]; (2) Target parameter pruning (TPP) [15], which prunes weights with low magnitude; (3) Random neuron pruning (RNP) [42]; and (4) Target neuron pruning (TNP) [15], which prunes neurons with low feature norm. For the comparison methods, we try different pruning percentages and report the best results. The ablation results in Table. 4 reveal that: (1) pruning

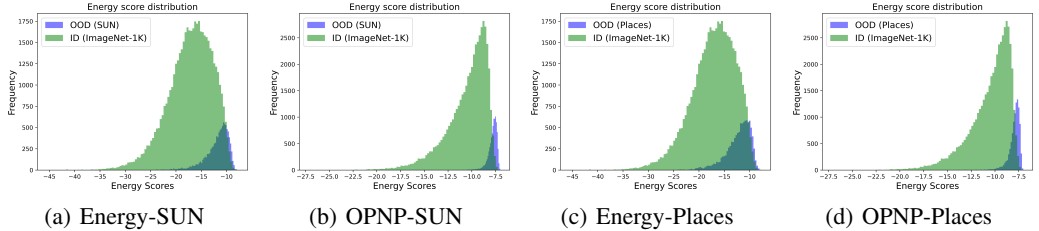

| (a) Energy-SUN | (b) OPNP-SUN | (c) Energy-Places | (d) OPNP-Places |

Figure 5: Illustration of OOD score distributions in two tasks with the Energy baseline and our proposed OPNP. (a) Energy baseline in SUN benchmark. (b) OPNP in SUN benchmark. (c) Energy baseline in Places benchmark. (d) OPNP in Places benchmark.

parameters and neurons with low magnitude outperforms random parameter and neuron pruning. (2) The introduced optimal parameter and neuron pruning outperforms other pruning methods by a margin margin, with 16.68% improvement in SUN dataset and 18.95% in Places dataset.

**OPNP benefits model calibration.** A well calibrated model should have better OOD detection performance [16]. In this ablation, we explore how OPNP influences model calibration. In Fig. 4, we evaluate model calibration performance with Confidence Reliability Diagrams (CRD) and Expected Calibration Error (ECE), which was introduced in [16]. As observed in Fig. 4(b), for an uncalibrated model, the confidence obviously exceeds accuracy, which indicates overconfident confidence. Fig. 4(c) illustrates the effect of utilizing OPNP, which shows that the consistency between confidence and accuracy is improved, and the ECE is reduced by 1.5%. It demonstrates that OPNP is beneficial to model calibration, which also explains why OPNP improves OOD detection performance.

**How OPNP changes the score distribution** In Fig. 5, we illustrate the OOD score distribution with the Energy baseline [31] and our proposed OPNP. We utilize SUN and Places benchmarks with ResNet50 model to exhibit how the OPNP changes the OOD score distributions. From the illustration, several interesting observations are: (1) Utilizing OPNP increases OOD scores for both ID and OOD samples. (2) The utilization of OPNP has a significant impact on the score distribution of OOD samples, resulting in a more condensed distribution. (3) The OOD score distributions of ID and OOD samples become more separable after applying OPNP, which validates the effectiveness of optimal parameter and neuron pruning.

## 5    Conclusion

**Conclusion and future work.** In this paper, we propose a simple yet effective post-hoc OOD detection method. In particular, a gradient-based method is proposed to estimate the sensitivity of model parameters and neurons. We show that the OOD detection performance could be significantly improved by simply removing the connection weights and neurons with exceptionally large or close to zero sensitivities. Extensive experiments and ablations are performed to demonstrate the effectiveness of our proposal. Compared to energy score baseline, our OPNP reduces FPR95 by 32.5% in ImageNet-1K benchmark. Besides, the OPNP outperforms the SOTA feature rectification method by 5.5% in FPR95 and outperforms the SOTA weight pruning method by 8.8% in FPR95. We believe the optimal parameter and neuron pruning is a promising direction for OOD detection tasks, and hope our findings can bring new ideas and breakthroughs to other researchers. In our future work, we will explore other post-hoc model pruning and quantization method, as well as low rank decomposition of model parameters for OOD detection.

**Limitation and societal impact.** The main limitation of this work is lack of theoretical guarantee. Therefore, we call for further application and explanation of the sensitivity guided parameter and neuron pruning method for OOD detection. This work aims to improve the safety of modern deep learning models, which tends to benefit a wide range of applications in social life, such as AI for medical, smart city and driverless system. We hope to provide a plug-and-play tool for AI model users to reduce the false recognition caused by OOD samples in the real world.

## Acknowledgments and Disclosure of Funding

This work was supported by the National Key R&D Program of China under Grant 2020AAA0103902.

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

# A  Evaluation on CIFAR Benchmarks

## A.1  Experimental setup

We additionally evaluate our proposal on the widely used CIFAR benchmarks with CIFAR10 and CIFAR100 as ID datasets. The standard split with 50,000 training ID images and 10,000 test ID images are utilized for training and evaluation. We following [43, 44] to evaluate on six common OOD datasets [4], including iSUN [50], LSUN-Resize [52], LSUN-Crop [52], SVHN [36], Places365 [56], and Textures [4]. Please refer to [44] for the details of the datasets. For both CIFAR10 and CIFAR100 datasets, we train a standard ResNet50 [18] model for 100 epochs. The learning rate is set to 0.01 and cosine annealing [32] is utilized for learning rate decay. The final accuracy of CIFAR10 and CIFAR100 classification model is 96.44% and 85.27% respectively.

## A.2  Main results

The main results on CIFAR10 and CIFAR100 benchmarks are show in Table 5 and Table 6. As can be seen, we consider the three most commonly used post-hoc methods and compare the OOD detection performance with and without the applying OPNP. The results show that: (1) On both CIFAR10 and CIFAR100, using OPNP consistently outperforms the counterpart without OPNP, which indicates that our proposal is compatible with other post-hoc methods. (2) The performance improvement brought by OPNP on CIFAR100 is more significant than on CIFAR10 and less significant than on ImageNet benchmark. We believe this is because the output layer on CIFAR10 classification model, which has fewer connection weights, is much less overparameterized than CIFAR100 and ImageNet classification models. (3) The OPNP achieves similar performance as ReAct On CIFAR10 benchmark, and outperforms ReAct on CIFAR100 benchmark. Besides, utilizing OPNP and ReAct jointly reduces FPR95 by 1.55% and 2.49% in CIFAR10 and CIFAR100 benchmark, respectively.

Table 5: OOD detection results on CIFAR10 benchmark with ResNet50 model. All numbers are percentages. The performances are reported as FPR95/AUROC.

| Method | OOD Datasets | | | | | | Average |
| | iSUN | LSUN-C | LSUN-R | SVHN | Places365 | Textures | |
|---|---|---|---|---|---|---|---|
| MSP[12] | 18.54/93.80 | 11.40/96.52 | 15.75/94.25 | 6.81/98.07 | 24.47/92.10 | 18.54/93.77 | 15.92/94.75 |
| **MSP+OPNP** | 11.52/95.44 | 7.27/98.49 | 12.59/96.10 | 3.32/98.87 | 19.41/94.40 | 14.84/96.10 | 11.49/96.57 |
| Energy [31] | 9.44/97.60 | 4.29/99.04 | 7.77/97.79 | 1.50/99.67 | 18.60/95.33 | 13.22/96.98 | 9.14/97.74 |
| **Energy+OPNP** | 7.24/98.18 | 3.91/99.04 | 6.58/98.45 | 0.86/99.82 | 13.65/96.73 | 7.40/98.31 | 6.61/98.42 |
| ReAct [43] | 8.81/97.94 | 4.94/98.92 | 6.29/98.34 | 2.31/99.48 | 11.24/97.31 | 5.93/98.48 | 6.59/98.41 |
| **ReAct+OPNP** | 5.54/98.32 | 3.62/99.25 | 4.48/99.00 | 0.40/99.90 | 9.34/97.64 | 6.88/98.20 | 5.04/98.72 |

Table 6: OOD detection results on CIFAR100 benchmark with ResNet50 model. All numbers are percentages. The performances are reported as FPR95/AUROC.

| Method | OOD Datasets | | | | | | Average |
| | iSUN | LSUN-C | LSUN-R | SVHN | Places365 | Textures | |
|---|---|---|---|---|---|---|---|
| MSP[12] | 51.22/83.16 | 49.73/86.08 | 45.77/84.78 | 31.85/91.48 | 40.76/87.17 | 40.31/87.79 | 43.27/86.74 |
| **MSP+OPNP** | 42.63/86.20 | 43.96/87.13 | 38.44/88.67 | 24.65/92.94 | 33.80/91.04 | 31.68/91.23 | 35.86/89.54 |
| Energy [31] | 38.39/89.72 | 30.77/93.18 | 32.00/91.54 | 15.21/96.94 | 32.96/93.06 | 25.43/93.56 | 29.13/93.00 |
| **Energy+OPNP** | 32.39/91.86 | 26.24/94.36 | 27.19/94.36 | 11.23/97.88 | 27.03/94.38 | 23.96/95.10 | 24.67/94.66 |
| ReAct [43] | 30.53/92.00 | 30.86/93.10 | 31.15/92.57 | 14.69/97.15 | 28.04/94.01 | 21.03/95.27 | 26.05/94.02 |
| **ReAct+OPNP** | 28.78/92.21 | 26.42/94.14 | 29.42/93.84 | 16.71/96.68 | 22.71/95.07 | 17.36/96.38 | 23.56/94.71 |

# B  More Ablation Studies

## B.1  Perform OPNP on different layers

In our main experiments, we apply the OPNP in the last FC layer in both ResNet50 and ViT-B/16. In this ablation, we also utilize the proposed OPNP in different intermediate layers of ResNet50.

---

[4]https://github.com/deeplearning-wisc/dice

As illustrated in Table. 7, we utilize the OPNP in four different convolutional layers, Layer1-Layer4, which represent the kernel weights of the last $1 \times 1$ convolutional layer in four *ResBlocks* of ResNet50. The results show that utilizing the OPNP in the last FC layer significantly outperforms the performance by using OPNP in the intermediate convolutional layers, which indicates that the overconfident predictions mainly comes from the last fully connected layer. Besides, we also estimate the parameter sensitivity of the whole model and pruning the weights with two globally pruning strategies, including (1) Global Threshold Pruning: pruning all the weights in Convolution layer and FC layer in a global threshold. (2) Layer-wise Pruning: pruning the weights in different layer individually with the same pruning ratio. We find that both global pruning methods perform worse than only pruning the FC layer. To help understand what happens when weights are pruned globally, we illustrate the average parameter sensitivity of different layers in Fig. 6. It shows that the sensitivity magnitude across different layers differs greatly, the sensitivity of the FC layer is less than $\frac{1}{10}$ of the shallow Convolution layer. Therefore, it is unreasonable to utilize a global threshold for all layers. It might be work if we use different thresholds for different layers, but it is too tricky to determine the optimal thresholds.

Table 7: Ablation study of applying OPNP on different layers of ResNet50. All numbers are percentages. The performances are reported as FPR95/AUROC.

| Where to OPNP | SUN | Places |
|---|---|---|
| Layers1 | 54.98/85.44 | 59.63/83.91 |
| Layers2 | 54.30/85.78 | 58.77/84.10 |
| Layers3 | 46.51/86.55 | 53.12/85.33 |
| Layers4 | 37.06/88.63 | 48.45/87.16 |
| Global Threshold Pruning | 39.74/91.91 | 49.13/88.32 |
| Layer-wise Pruning | 40.48/92.35 | 44.92/90.27 |
| **FC Layer** | 18.50/95.62 | 30.14/93.46 |

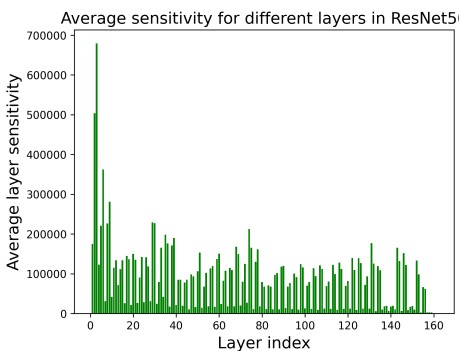

Figure 6: Average parameter sensitivity of different layers in ResNet50.

## B.2 Parameter sensitivity estimation with different number of training samples.

To reduce the cost for parameter sensitivity estimation, it is advisable to compute the parameter sensitivity based on a subset of the training set rather than the entire set. To demonstrate the effectiveness of using a subset for sensitivity estimation. We randomly select 1%, 5%, 20% and 100% training samples for sensitivity estimation, and perform parameter pruning based on the sensitivities. The experimental results are shown in Table 8, which demonstrate that using only 1% of the training set (ImageNet-1K) also achieves promising performance.

## B.3 Neuron sensitivity estimation with different statistics

In the main experiments, we utilize the average sensitivity (equivalent to $\ell_1$ norm ) as the neuron sensitivity. It is worth noting that other statistics, such as $\ell_2$ norm, Variance, Maximum (Max),

Table 8: Comparison of experimental results when estimating parameter sensitivity with different numbers of training samples. The performance of OPP in ResNet50 is reported. w/o indicates Energy baseline without parameter pruning. Numbers are percentages.

| Sampling Ratio | w/o pruning | 1% | 5% | 20% | 100% |
|---|---|---|---|---|---|
| SUN | 59.3/85.9 | 32.57/92.83 | 32.06/92.78 | 30.92/93.05 | 30.40/93.17 |
| Places | 64.9/82.9 | 42.30/90.00 | 41.96/90.08 | 41.21/90.33 | 40.76/90.65 |

Table 9: Comparison of experimental results when estimating neuron sensitivity with different statistics. The performance in ResNet50 is reported. Numbers are percentages.

| Statistics | Mean | Max | Min | Median | $\ell_2$ Norm | Variance |
|---|---|---|---|---|---|---|
| SUN | 26.7/94.7 | 32.4/93.1 | 26.0/95.1 | 26.3/94.6 | 31.2/92.7 | 46.1/88.5 |
| Places | 32.7/92.9 | 39.8/91.2 | 32.6/92.8 | 33.4/92.5 | 34.5/91.2 | 51.1/87.4 |

Minimum (Min) and Median may also feasible statistics to measure the neuron sensitivity. Therefore, we compare the performance of using different statistics as the neuron sensitivity. The results are presented in Figure 9, which show that using the Mean, Min and Median of the weights sensitivity as neuron sensitivity achieve similar performance, and using the Variance statistic performs worst. As the Mean value is more robust and less likely to be susceptible by noisy connections, we suggest the users utilize the Mean statistic of parameter sensitivity for neuron pruning.

## C   Sensitivity Distribution Visualization

### C.1   Parameter and neuron sensitivity distribution on CIFAR benchmarks.

In Fig. 7 and Fig. 8, we illustrate the parameter and neuron sensitivity distribution for CIFAR10 and CIFAR100 classification models, which reveal several interesting insights. First, on both CIFAR10 and CIFAR100 classification models, there are a few parameters that exhibit exceptionally high sensitivities and close to zero sensitivities, which is consistent with the sensitivity distribution on the ImageNet benchmark. Second, there are more parameters with close to zero sensitivities and more neurons with abnormal sensitivities on CIFAR100 classification model, which explains why OPNP brings more performance gains on CIFAR100 benchmark.

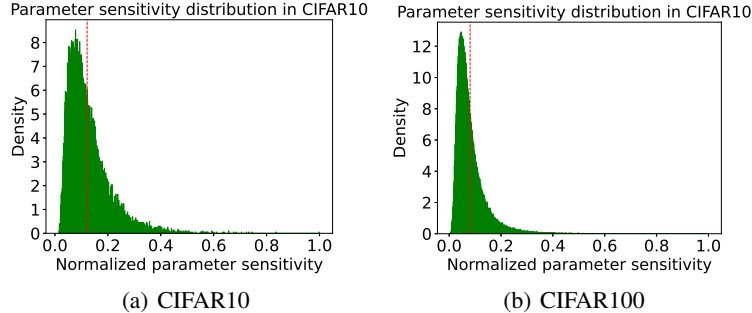

(a) CIFAR10        (b) CIFAR100

Figure 7: Illustration of parameter sensitivity distribution of classification models on (a) CIFAR10 and (b) CIFAR100. The parameters are taken from the last fully-connected layer in ResNet50, which are $10 \times 2048$ and $100 \times 2048$ for CIFAR10 and CIFAR100 classification model respectively. The dotted line in red indicates the average sensitivity and the maximum sensitivity is normalized to 1.

### C.2   Parameter Sensitivity Distribution of the Pruned Weights

The parameter sensitivity distribution (on ID and OOD set) of the pruned weights is illustrated in Fig. 9. The results show that the pruned weights contain many high sensitive connections for OOD set.

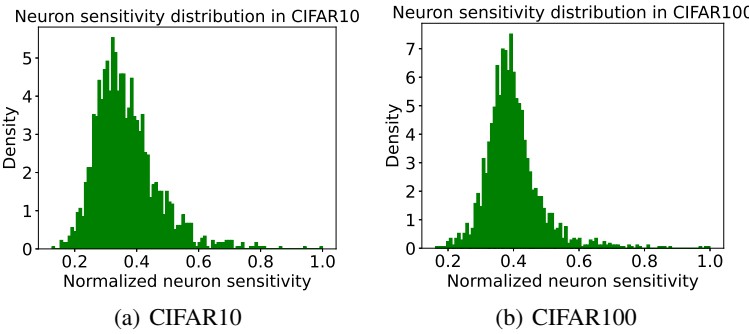

|  (a) CIFAR10 | (b) CIFAR100 |

Figure 8: Illustration of neuron sensitivity distribution on (a) CIFAR10 and (b) CIFAR100. The neurons are from the pre-logits layer in ResNet50. The maximum sensitivity is normalized to 1.

Besides, the average sensitivity on OOD set (0.00024) is larger than the average sensitivity on ID set (0.00018), which experimentally verifies Eq. 11.

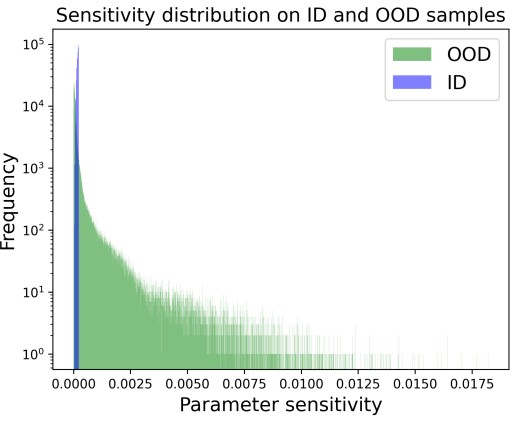

Figure 9: Sensitivity distribution on ID and OOD samples. The top 20% sensitive parameters (measure on ID set) are illustrated. The average sensitivity on OOD samples is 0.00024 and the average sensitivity on ID samples is 0.00018.

## D    Discussion

The whole OOD detection pipeline is presented in Algorithm 1. In the main experiments, we integrate parameter pruning and neuron pruning in a unified framework, which actually can be used individually. The effectiveness of parameter and neuron pruning may vary for different model architectures and ID tasks. Therefore, we recommend readers to attempt different pruning percentage combinations for different tasks. Energy score [31], ReAct [43], DICE [44] ans ASH [6] are the most relevant researches to our work. The difference between our OPNP and the baseline method - Energy Score [31] is that we remove some output connections and hidden neurons according to pre-computed parameter and neuron sensitivities. Different from ReAct [43], which clips features with abnormal magnitude, our OPNP removes the parameters and neurons with abnormal sensitivity. The differences between our method and DICE [44] are: (1) DICE obtains the weight importance by averaging the contribution of weights to logit output, while our method measures the parameter and neuron sensitivity to energy score by averaging gradient; (2) DICE only removes the weights with low importance, while our OPNP not only removes the non-sensitive connections and neurons but also remove the most sensitive ones. Besides, ASH clips or binarizes the features based on the magnitude to improve OOD performance, whereas our OPNP prunes both weights and neurons based on sensitivity. Compared to those methods that also explore sparsity to improve OOD detection performance, the

introduced sensitivity metric measured by output energy is directly coupled with OOD detection tasks, and the sensitivity guided pruning is more intuitive, comprehensive and user-friendly.

---

**Algorithm 1** OPNP - Optimal Parameter and Neuron Pruning

---

**Input:** A trained model $f(\boldsymbol{x}; \boldsymbol{\theta})$, training samples $\{\boldsymbol{x}_i\}_{i=1}^m$, pruning percentage $\rho_{max}^w, \rho_{min}^w, \rho_{min}^o$, $\rho_{max}^o$, test sample $\{\boldsymbol{x}_k\}_{k=1}^n$;

**Output:** Energy score of test sample $\{E(\boldsymbol{x}_k)\}_{k=1}^n$

1: **for** $i = 1$ to $m$ **do**
2:      compute gradient $g(\boldsymbol{x}_i)$;
3: **end for**
4: $\mathbf{M} = \frac{1}{m} \sum_{k=1}^m |g(\boldsymbol{x}_k)|$ ;
5: $\mathbf{O}_i = \frac{1}{K} \sum_{p=1}^K \mathbf{M}_{ip}, \quad i = 1, \cdots, L$;
6: Get threshold $\Omega_{min}^w, \Omega_{max}^w$ by ranking $\mathbf{M}$;
7: Get threshold $\Omega_{min}^o, \Omega_{max}^o$ by ranking $\mathbf{O}$;
8: $\mathbf{W}[\mathbf{M} > \Omega_{max}^w] = 0$ and $\mathbf{W}[\mathbf{M} < \Omega_{min}^w] = 0$;
9: **for** $k = 1$ to $n$ **do**
10:      Compute the pre-logit embedding $h(\boldsymbol{x}_k)$;
11:      $h(\boldsymbol{x}_k)[\mathbf{O} > \Omega_{max}^o] = 0$ and $h(\boldsymbol{x}_k)[\mathbf{O} < \Omega_{min}^o] = 0$;
12:      $f(\boldsymbol{x}_k) = \mathbf{W} \cdot h(\boldsymbol{x}_k) + \boldsymbol{b}$;
13:      $E(\boldsymbol{x}_k) = -\log \sum_{i=1}^K \exp(f_i(\boldsymbol{x}_k))$;
14: **end for**
15: **Return** $\{E(\boldsymbol{x}_k)\}_{k=1}^n$

---

