# OpenReview forum: "Optimal Parameter and Neuron Pruning for Out-of-Distribution Detection"
_NeurIPS.cc/2023/Conference — NeurIPS 2023 poster_

### Official Review · Reviewer_qR61 · 2023-06-25

**Soundness:** 3 good
**Presentation:** 4 excellent
**Contribution:** 2 fair
**Rating:** 6
**Confidence:** 4

**Summary:**

This paper contributes a new parameter and neuron pruning methods for OOD detection. Built upon the energy-based score, this paper defines the sensitivity of a parameter (neuron) wrt the energy-score by using gradient.

**Strengths:**

The arguments are clear and easily understood. The method is well motivated by removing the weights and neurons. The experiments are comprehensive and clear, and the results are promising.

**Weaknesses:**

$\bullet$ The major weakness is the theoretical explanation between the sensitivity and OOD performance, but this is clearly pointed out in the paper.

$\bullet$ The usage of the sensitivity is based on intuition, while this is a good technique for solving OOD problems.

**Questions:**

$\bullet$ What are the techniques (feature ensemble or/and input preprocessing) used in Mahalanbis score in comparison?

$\bullet$ This pruning method seems to be very promising. I understand this was discussed as limitation and future work, but is it possible to share the insights why the sensitivity term(s) will work?

**Limitations:**

The limitations are fully discussed with explanations.

---

> ### Author Rebuttal · Authors · 2023-08-09
>
> **Comment** We thank Reviewer qR61 (R4) for the helpful suggestions.
>
> > **W:** The major weakness is the theoretical explanation between the sensitivity and OOD performance, but this is clearly pointed out in the paper. The usage of the sensitivity is based on intuition, while this is a good technique for solving OOD problems.
>
> **A:** We provide three remarks to explain the insight of our work
>
> ## Insight Justification
> **Remark 1: Parameter and neuron pruning avoid overconfident predictions**
>
> The over-parameterized deep neural networks always generate overconfident predictions even for OOD samples [1,2,3]. Therefore, most exsiting methods improve OOD performance by avoiding overconfident predictions [4,5]. For a deep network, the last fully connected layer can be regarded as a linear classifier. To prevent the classifier from overfitting, the most widely used technique is $\ell_1$ and $\ell_2$ regularization,  which can be formulated as
>
> $\min_{\theta}\mathbb{E}_{(x,y)\sim D}\Vert\theta^\top\cdot h(x)-y\Vert_2^2 + \lambda\mathcal R(\theta)$
>
> Here $\mathcal{R}(\theta)$ represents $\ell_1\text-$ or $\ell_2\text{-norm}$ of $\theta$. As sensitivity based parameter pruning is able to reduce $\mathcal{R}(\theta)$,  it can be regarded as an effective post reguralization technique that reduces the model complexity and avoids overconfident outputs. Besides, the neuron pruning is similar to dropout technique [6] which has also been demonstrated to reduce overfitting. Therefore, the proposed OPNP can avoid overconfident predictions and potentially improve the OOD detection performance.
>
> **Remark 2: Pruning the least sensitive parameters and neurons improve separability between ID and OOD samples**
>
>  We denote $f_j(x)$ the logit output of $j\text{-th}$ class, denote $\mathbf{W}$ the output weights, denotes $\mathbf{M}$ the sensitive matrix of $\mathbf{W}$. After pruning the least sensitive parameters, the logit reduction of the j-th class can be estimated as
>
> $\Delta f_j(x) = \sum_{\mathbf{M}\_{jk}<\Omega\_{min}^w}\mathbf{M}\_{jk}\cdot\lvert\mathbf W_{jk}\rvert\cdot h_k(x)$
>
> where $L$ denotes the number of hidden neurons, and $\Omega_{min}^w$ denotes the sensitivity threshold. Eq.2 shows that the logit reduction is positively correlated with the average sensitivity of the pruned weights. As the parameter sensitivity is computed over the training ID set, the least sensitive parameters on ID distribution should be more sensitive for OOD samples on average, i.e.,
>
> $\sum_{\mathbf{M}\_{jk}<\Omega\_{min}^w}\mathbf{M}\_{jk}^{OOD} > \sum_{\mathbf{M}\_{jk}<\Omega\_{min}^w} \mathbf{M}_{jk}^{ID}$
>
> Therefore, the logit reduction on OOD samples is larger than on ID samples, which tends to improve the separability between ID and OOD samples, and leads to better OOD detection performance.
>
> We also show the parameter sensitivity distribution (on ID and OOD set) of the pruned weights in Fig. 1 (see the PDF), which demonstrates that: (1) The pruned weights contain many high sensitive connections for OOD set. (2) The average sensitivity  (0.00024) on OOD set is larger than the average sensitivity (0.00018) on ID set. This experimentally verifies Eq. 3.
>
> **Remark 3: Pruning the most sensitive parameters and neurons improves generalization**
> We follow [10] to define the first-order flatness as
>
> $R_\rho(\theta) \triangleq \rho\cdot\max_{\theta'\in B(\theta,\rho)}\Vert\Delta f(\theta')\Vert, \quad \forall\theta\in\Theta$
>
> where $\Delta f(\theta')$ denotes the derivative at point $\theta$, $B(\theta,\rho) = {\theta': \Vert\theta-\theta'\Vert<\rho}$ denotes the open ball of radius $\rho$ centered at the point $\theta$ in the Euclidean space and $\rho$denotes the perturbation radius that controls the magnitude of the neighbourhood. The flatness $R_\rho(\theta)$ descripts how flat the function landscape is [9]. It has been demonstrated that a flatter landscape could lead to better generalization [7,8,9].  Eq. 4 indicates that the first-order flatness is determined by the largest gradient. Therefore, a smaller gradient norm leads to flatter landscape, and many recent works have been proposed to penalize the gradient norm to get better generalization performance [7,8]. Our proposed method pruning the most sensitive parameters, therefore, is able to improve the flatness of the function landscape and lead to better generalization. However, according to Remark 2, pruning the most sensitive parameters and neurons may also hurts the separability between ID and OOD samples. Therefore, there is a trade-off between better generalization and better ID-OOD separability. This explains why OOD performance improves with very few sensitive parameters pruned and drops with a large pruning ratio (see in Fig. 3 of our paper ).
>
>
> > **Q:** What are the techniques (feature ensemble or/and input preprocessing) used in Mahalanbis score in comparison?
>
> **A:**  For the results of Mahalanbis score, we follow [1,2] to report the performance in their paper since we utilize the same evaluation setting. The Mean and Covariance matrix are computed based on the training set. The implementation details of the Mahalanbis score can be seen in [3]
>
> [1] React: Out-of-distribution detection with rectified activations. NeurIPS 2021.
>
> [2] DICE: Leveraging Sparsification for Out-of-Distribution Detection. ECCV 2022.
>
> [3] A simple unified framework for detecting out-of-distribution samples and adversarial attacks. NeurIPS 2018.
>
> [4] Energy-based Out-of-distribution Detection. NeurIPS 2020.
>
> [5] DICE: Leveraging Sparsification for Out-of-Distribution Detection. ECCV 2022
>
> [6] Learning sparse networks using targeted dropout. Arxiv 2019.
>
> [7] Sharpness-aware minimization for efficiently improving generalization. ICLR 2021
>
> [8] Penalizing Gradient Norm for Efficiently Improving Generalization in Deep Learning. ICML 2022
>
> [9] Gradient Norm Aware Minimization Seeks First-Order Flatness and Improves Generalization. CVPR 2023

---

> > ### Comment · Reviewer_qR61 · 2023-08-11
> > **Thank you for your reply**
> >
> > Thank you for your reply. I like the explanation. Thank you again and good luck.

---

> > > ### Author Response · Authors · 2023-08-12
> > >
> > > Thank you again for your further review and immediate feedback. We are happy to provide more explanations if you have any other questions.

---

### Official Review · Reviewer_tCft · 2023-07-03

**Soundness:** 4 excellent
**Presentation:** 3 good
**Contribution:** 4 excellent
**Rating:** 6
**Confidence:** 4

**Summary:**

This paper proposes a parameter and neuron pruning strategy to enhance out-of-distribution detection. The approach involves removing near-zero- and high-sensitivity parameters, which are measured by the average gradient corresponding to all training in-distribution samples. Empirical results demonstrate the superior performance of the proposed algorithm.

**Strengths:**

The proposed algorithm is characterized by its simplicity and remarkable effectiveness in out-of-distribution detection tasks, exhibiting consistently high performance.


**Weaknesses:**


While the proposed principle shows promising results, providing theoretical explanations for its success would be valuable. Additionally, selecting the appropriate hyper-parameters beforehand presents challenges.


**Questions:**

Although empirical evidence indicates performance gains when pruning the largest sensitivity value (as seen in Figure 3), understanding its behavior and identifying optimal values require further investigation. Users would benefit from discussions clarifying such cases and receiving intuitive suggestions to aid in parameter selection. Additionally, exploring the impact of using a fixed threshold instead of a percentile for pruning sensitivity values may lead to improved performance.

The reliance on the average sensitivity of connection weights for neuron pruning lacks an intuitive explanation. Other statistics, such as min, max, or median, could also be considered as potential alternatives.

An explicit explanation of how to combine OPNP with ReAct should be provided.

To enhance efficiency, it is advisable to consider pruning based on a subset of the training set rather than the entire set.


**Limitations:**

The authors have addressed some limitations of their work, and there are additional suggestions for improvement in the 'Paper Weakness'' part and "Questions" part.

---

> ### Author Rebuttal · Authors · 2023-08-09
>
> **Comment** We thank Reviewer tCft (R3) for the insightful questions and suggestions, which really helped us improve our paper. Here, we respond to the questions and suggestions point by point.
>
> > **W1:** While the proposed principle shows promising results, providing theoretical explanations for its success would be valuable.  Users would benefit from discussions clarifying such cases and receiving intuitive suggestions to aid in parameter selection
>
> ***
>
> **A:** Thanks for your suggestion. We provided three remarks to explain the insight and explanations of why the gradient based sensitivity works. Please refer to the response to **Review qR61** for the details of the remarks.
>
> **Remark 1: Parameter and neuron pruning avoid overconfident predictions**
>
> **Remark 2: Pruning the least sensitive parameters and neurons improve separability between ID and OOD samples**
>
> **Remark 3: Pruning the most sensitive parameters and neurons improves generalization**
>
> ***
>
> > **Q1:** Selecting the appropriate hyper-parameters beforehand presents challenges. Exploring the impact of using a fixed threshold may lead to improved performance.
>
> **A:** As we don't know the distribution of sensitivities, it is difficult to determine a fixed threshold. We refer to ReAct [3] and DICE [4] to use a prunning ratio, and explore the ratio on the validation set. The results in Tabel 1 and 2 show that the selected pruning ratio works for different OOD sets.
>
>
> ***
> > **Q2:** The reliance on the average sensitivity of connection weights for neuron pruning lacks an intuitive explanation. Other statistics, such as min, max, or median, could also be considered as potential alternatives.
>
> **A:**  It is difficult to prove theoretically which statistic works best. When we design the sensitivity metric for neuron pruning, we have tried different statistics, such as mean,  variance and norm, the results showed that average works best.
>
> For the $i\text{-th}$ neuron in the penultimate layer, it contributes to all output neurons, therefore the sensitivity of the $i\text{-th}$ neuron should be defined based on the sensitivity of the connection weights between the $i\text{-th}$ hidden neuron and all output neurons. Considering that the $\ell_1$ or $\ell_2$ norm of the weights are usually used to meansure the importance of the units in deep networks [1, 2]. We define the neuron sensitivity as the average sensitivity of the connections, which is equivalent to to using $\ell_1$ norm of the weight sensitivity, and reflects the average sensitivity to all classes.
>
> **Ablation study on different statistics**
>
> In the main experments, we utilize the average sensitivity (equivalent to $\ell_1$norm ) as the neuron sensitivity. It is worth noting that other statistics, such as $\ell_2$ norm, variance, max, min, median may also a feasible statistics to measure the neuron sensitivity. Therefore, we perform a comparision experiment to show the performence of using different statistics as the neuron sensitivity. The results are presented as follows, which show that using the Mean, Min and Median of the weights sensitivity as neuron sensitivity achieve similar performance, and using the Variance statistic performs worst. As the Mean value is more robust and less likely to be susceptible by noisy connections, we suggest the users utilize the Mean sensitivity for neuron pruning.
>
> FPR95 and AUROC  in SUN and Places datasets are reported, ResNet50 model is utilized.
>
> | Statistics |    Mean   |    Max    |     Min    |    Median   |    Norm   |   Variance  |
> |:----------:|:---------:|:---------:|:----------:|:-----------:|:---------:|:-----------:|
> |     SUN    | 26.7/94.7 | 32.4/93.1 |  26.0/95.1 |  26.3/94.6  | 31.2/92.7 |  46.1/88.5  |
> |   Places   | 32.7/92.9 | 39.8/91.2 | 32.6/92.8 |   33.4/92.5 | 34.5/91.2 |   51.1/87.4 |
>
>
> We will revise the submission (lines 172-174) to add more intuitive explanation and discuss the results of using different statistics for neuron pruning in the appendix.
>
> ***
>
> > **Q3:** An explicit explanation of how to combine OPNP with ReAct should be provided.
>
> **A:** Thanks for your suggestion. ReAct is a clipping operation that truncates activations above c to limit the effect of noise. It can be defined as $\overline h(x) = \min(h(x), c)$, where $c$ is the clipping threshold.  Therefore, ReAct can be integrated into our method easily by clipping the neuron outputs after our neuron pruning stage. The clipping ratio is set to 10% in ResNet50 and 5% in ViT-B/16.
>
>  We will add the information in the revised paper as suggested.
>
> ***
> > **Q4:** To enhance efficiency, it is advisable to consider pruning based on a subset of the training set rather than the entire set.
>
> **A:** Good suggestion. That is actually what we do when we try different strategies to get the parameter sensitivity.
>  We performed comparision experiments by using different number of training samples for parameter sensitivity estimation. The results are presented as follows, which shows that estimating the sensitivity with only 1% training samples also achieved promising performance. The ablation results will be added to the appendix of our paper.
>
>
> FPR95 and AUROC  in SUN and Places datasets are reported, ResNet50 model is utilized. w/o pruning denotes baseline without pruning.
>
> | Sampling Ratio |       w/o pruning | 1%          | 5%          | 20%         | 100%        |
> |----------------|-------------------|-------------|-------------|-------------|-------------|
> | SUN            | 59.3/85.9         | 32.57/92.83 | 32.06/92.78 | 30.92/93.05 | 30.40/93.17 |
> | Places         | 64.9/82.9         | 42.30/90.00 | 41.96/90.08 | 41.21/90.33 | 40.76/90.65 |
>
> ***
>
>  **Reference**
>
> [1] Pruning Filters For Efficient ConvNets. ICLR 2017.
>
> [2] Learning sparse networks using targeted dropout. Arxiv 2019.
>
> [3] React: Out-of-distribution detection with rectified activations. NeurIPS 2021.
>
> [4] DICE: Leveraging Sparsification for Out-of-Distribution Detection. ECCV 2022.

---

> > ### Comment · Reviewer_tCft · 2023-08-18
> > **Response to Author Rebuttal**
> >
> > Thank you for your rebuttal; it has effectively addressed most of my concerns.

---

### Official Review · Reviewer_8qxy · 2023-07-06

**Soundness:** 2 fair
**Presentation:** 2 fair
**Contribution:** 2 fair
**Rating:** 5
**Confidence:** 3

**Summary:**

This submission proposes a post-hoc method for detecting out-of-distribution samples, by pruning the final classification layer, using a sensitivity metric based on the gradients of the energy scores. The proposed method is mainly validated on residual networks and visual transformers based on the ImageNet dataset.


**Strengths:**

- The submission proposes an interesting application using pruning to out-of-distribution detection, the latter being an important research problem
- The proposed method is simple, and can be applied with a fairly small computational overhead
- The results suggest that the proposed method, also when used in conjunction with other methods, can achieve state-of-the-art results on several tasks.



**Weaknesses:**

- The submission is not very well written, particularly the introduction, with quite a few typos and grammar issues (e.g. using “post-hot” instead of “post-hoc” in quite a few places). This makes reading a bit difficult.
- The proposed method is not very novel, as other OOD detection methods have used pruning (e.g. references 20, 35 seem quite similar)
- The performance of the method depends quite heavily on the pruning thresholds (see Figure 3) and can substantially decrease the accuracy on the original ID classification task (see Table 3). This makes the method less likely to be used “out of the box”.
- While the authors introduce their method as "post-hoc", in fact it has a similar cost to training the model for one epoch, since the gradients for all the samples have to be computed.


**Questions:**

- Suggestion: I would advise the authors to go through the submission and correct the typos and grammar issues, for the next revision
- I believe Equation (4) is an approximation, rather than an identity. Have the authors considered what would be a theoretical explanation that would connect the approximation in Equation (4) and the removal of least and most sensitive weights?
- What is the norm in Equation (6)? Should it be an absolute value instead?
- Can the authors provide more explanations on the intuition that “parameters and neurons with exceptionally large sensitivity can easily lead to overconfidence” (51-52)?
- Lines 145-147: it is hard to see these differences from Figure 1, I would suggest adding numbers
- Lines 181-182: this is not really a Gaussian (e.g it is not symmetric)
- Can you provide more details on how OPNP is combined with ReAct?
- In Table 2, can you provide results for DICE + ReAct?
- Some of the plots and tables do not include information about the model and dataset used. For example, what is the model used in Table 3 or Table 4? What is the model and dataset from Figure 4?
- The authors show results when (independently) pruning other layers in a ResNet50 model (Appendix Table 3). What happens if the weights are pruned globally, instead of each layer at a time?
- Have the authors considered second order information for computing the sensitivity metric? For example, a second order Taylor approximation could be used instead of Equation (4), and since pruning is only done for the final layer, the Hessian could be more easily calculated. This is a similar approach to Optimal Brain Surgeon (Hassibi et al.)
- Can you clarify what is meant by "(subset of) iNaturalist, SUN, Place and Texture" (line 211)? Are the same corresponding datasets used for evaluation when comparing with the other methods from Tables 1 and 2?
- Can you please confirm that the ID test set used for validation (and determining the pruning hyperparameters and OOD threshold) was different than the one used for the final testing?
- Can you please clarify the differences between Algorithm 1 and the text which specifies that the parameter and neuron sensitivities are computed based on the entire training set? In line 5 from Algorithm 1, it sounds like neuron pruning is done individually for each sample, but from the text the neuron sensitivities are computed on the entire training set. In my understanding, the model is pruned only once, and the same sparse model is used for evaluation on different OOD tasks. Can you please confirm whether my understanding is correct?


**Limitations:**

While the authors have addressed some of the limitations of their work in Section 5, I believe the proposed method, in its current form, has some technical limitations that would make it more difficult to use “plug-and-play” as the authors hope. For example, the method relies quite heavily on the optimal hyperparameters (thresholds) for pruning and it can have a substantial negative impact on the ID classification accuracy for the original task. I would advise the authors to consider how the search of the pruning hyperparameters could be automatized, or at least made more efficient.

--------------------------------
**Edited after rebuttals**

After reading the authors' answer, I have raised my score from 4 to 5.

---

> ### Author Rebuttal · Authors · 2023-08-09
>
> **Comment** We thank Reviewer 8qxy (R2) for the careful reviews and insightful suggestions, which really helped us improve our paper. Due to space constraints, part of the responses can be seen in our global response.
>
> ***
>
> > **W1:**  Typos and grammar issues.
>
> **A:** We have corrected the typos in the revised paper, and will polish the whole paper carefully in the next version.
>
> ***
>
> >  **W2:** The proposed method is not very novel, as other OOD detection methods have used pruning (e.g. Ref 20, 35)
>
> **A:** In our global response, we further explained our insight through 3 remarks and explained the novelty of our proposal .
>
> ***
>
> > **W3:** The performance depends heavily on the thresholds (Fig.3)  and can substantially decrease the ID accuracy (Tab.3).
>
> **A:** (1) According to Fig.3, of all the hyperparameters, only parameter $\rho_{max}^w$ is relatively sensitive, which can be explained by **Remark 3** (See the global response). (2) The performance is consistently improved in a wide range of pruning ratio. The optimal pruning ratio can be set to $\rho_{min}^w\in[10,30]$ and $\rho_{max}^w\in[0.5,3]$ across different OOD set (see in Fig. 3).（3）As we only modified the last FC layer, we can always use the original FC layer for classification, which ensures an identical classification accuracy as unpruned model. Existing ReAct and DICE also rely on using the original FC layer for classification.
>
> ***
>
> >  **W4:** The authors introduce their method as "post-hoc",  but it has a similar cost to training the model for one epoch
>
> **A:** The "post-hoc" is relative to the training based method. The cost of parameter sensitivity estimation is much cheaper than training the model, since we only compute the gradient of the last FC layer, and the parameter sensitivity could be estimated in a subset of the training data (See the global response) . Therefore, the training cost is less than 1% compared to the training based methods (even for one epoch).
> Although our method is more costly than MSP and Energy, it has similar cost as other popular post-hoc methods (ReAct and DICE), while achieves better performance.
>
> ***
>
> > **Q1:** Equation (4) is an approximation, rather than an identity. Any theoretical explanation of removing the least and most sensitive weights?
>
> **A:** Thans for your seggestion, we have revised Eq. 4 as
>
> $E(x_k; \theta+\delta) - E(x_k; \theta) \approx \sum_{i,j}g_{ij}(x\_k)\delta_{ij}$
>
> We have provided three remarks in the global response to explain why sensitivity based pruning works.
>
> ***
> > **Q2:**  What is the norm in Equation (6)? Should it be an absolute value instead?
>
> **A:** Yes, it should be absolute, we have revised it as
>
> $\mathbf{M}\_{ij} = \frac{1}{m}\sum_\{k=1}^m\lvert g_{ij}(x_k)\lvert$
>
> ***
>
> > **Q3:** Provide more explanations on the intuition that “parameters and neurons with exceptionally large sensitivity can easily lead to overconfidence” (51-52)?
>
> **A:** We have provided three remarks to explain the insight in our global response. The function landscape is flat if the model output does not change drastically in the neighbourhood of the model parameter. Many existing works have shown that a flat landscape lead to better generalization. We have revised the sentence as "As the parameters with exceptionally large sensitivity can lead to a sharp landscape, which has been proved to hurts model generalization [1]"
>
> ***
>
> > **Q4:**  Lines 145-147: it is hard to see these differences from Figure 1, I suggest adding numbers
>
> **A:** We have revised Fig.1 as suggested.
>
> ***
>
> > **Q5:** Lines 181-182: this is not really a Gaussian
>
> **A:** The sentence has been revised as "As observed, before neuron pruning, there are several risky neurons which has typical large sensitivities."
>
> ***
>
> > **Q6:** More details on how OPNP is combined with ReAct?
>
> **A:** We have provided the details of OPNP+ReAct in the global response.
>
> ***
>
> > **Q7:** In Table 2, can you provide results for DICE + ReAct?
>
> **A:** We have added the results for DICE+ReAct in Table 2 as suggested. Results are as follows.
>
> FPR95/AUROC are reported.
>
> | OOD Dataset | iNaturalist |     SUN     |    Places   |   Texture   |   Average   |
> |:-----------:|:-----------:|:-----------:|:-----------:|:-----------:|:-----------:|
> |  DICE+ReAct |  2.65/99.38 | 29.45/93.52 | 38.45/91.17 | 33.78/93.27 | 26.08/94.34 |
>
> ***
>
> > **Q8:** Some of the plots (Fig. 4) and tables (Table 3, 4) do not include information about the model and dataset used.
>
> **A:**  ResNet50 model is used in both Table 3, Table 4 and Fig. 4. In Fig.4,  we utilize the prediciton results in ImageNet-1K to evaluate the calibration performance. We have added the information in the revised paper.
>
> ***
>
> > **Q9** What happens if the weights are pruned globally?
>
> **A:** Please see the **Ablations on global pruning** in our global response.
>
> ***
>
> > **Q10:** Have the authors considered second order information for computing the sensitivity?
>
> **A:** Yes, we have considered using the eigenvalues of the hessian matrix to measure parameter sensitivity. However, for the ResNet50 in ImageNet-1K, the hessian matrix is 2,048,000x2,048,000, which needs about 8TB Memroy in float16.
>
> ***
>
> > **Q11:** Can you clarify what is meant by "(subset of) iNaturalist, SUN, Place and Texture" (line 211)?  Please confirm that the ID test set used for validation was different than the one used for testing?
>
> **A:** Please see the **Evaluation Setting** in our global response.
>
> ***
>
> > **Q12:** In line 5 from Algorithm 1, the neuron pruning is done individually, but from the text the neuron sensitivities are computed on the entire training set.
>
> **A:** The pruned parameter and neuron index is computed once, which are obtained based on the training set.  But the neuron pruning for different test samples is done individually by setting the correspinding feature to zero in inference time. (See the revised pseudocode in PDF)
>
> [1] Sharpness-aware minimization for efficiently improving generalization

---

> ### Comment · Reviewer_8qxy · 2023-08-15
>
> Thank you for addressing my questions! After reading the answers and the other reviews, I have raised my score from 4 to 5.

---

> > ### Author Response · Authors · 2023-08-16
> >
> > We thank the reviewer again for evaluating our work and carefully reading our response. It is more than welcome to post any further comments.

---

### Official Review · Reviewer_Dntp · 2023-07-07

**Soundness:** 3 good
**Presentation:** 2 fair
**Contribution:** 2 fair
**Rating:** 5
**Confidence:** 4

**Summary:**

This paper proposes to adopt weight and neuron pruning for OOD detection. The proposed method is able to be combined with training-based approaches, demonstrating SOTA performance.

**Strengths:**

1. The motivation of the method is reasonable and sound.

2. The results of OPNP+ReAct is strong.

**Weaknesses:**

1. It is not new for the ML community that sparsity can help improve OOD detection. Numerous related works have been proposed to adopt sparsity/pruning to improve OOD detection. While might having different pruning approaches, it is necessary to discuss and compare them in the submission.

[1] Cheng, Zhen, et al. "Average of pruning: Improving performance and stability of out-of-distribution detection." arXiv preprint arXiv:2303.01201 (2023).

[2] Djurisic, Andrija, et al. "Extremely simple activation shaping for out-of-distribution detection." arXiv preprint arXiv:2209.09858 (2022).

[3] Sun, Yiyou, and Yixuan Li. "Dice: Leveraging sparsification for out-of-distribution detection." European Conference on Computer Vision. Cham: Springer Nature Switzerland, 2022.

[4] Liu, Shiwei, et al. "Deep ensembling with no overhead for either training or testing: The all-round blessings of dynamic sparsity." arXiv preprint arXiv:2106.14568 (2021).

2. The pseudocode of Algorithm 1 looks very trivial, and can be significantly improved.

**Questions:**

See the above limitation.

---

> ### Author Rebuttal · Authors · 2023-08-09
>
> **Comment**
> We thank Reviewer Dntp (R1) for the feedback and suggestions. Here, we respond to the concerns point by point.
>
> > **W1:**  It is not new for the ML community that sparsity can help improve OOD detection. Numerous related works have been proposed to adopt sparsity/pruning to improve OOD detection [1,2,3,4]. While might having different pruning approaches, it is necessary to discuss and compare them in the submission.
>
> **Ans:**  This is a good question and the recommended papers are related to our work, we will improve Section 3.4 according to the suggestion.
>
> In our global response, we provided three remarks to explain the insight of our work, and discussed the relationship/difference between our method and these related methods.
>
>  We acknowledge that sparsity and pruning are common strategies in machine learning, and some previous work have adopted sparsity/pruning to improve OOD detection [1,2,3,4]. However, Our proposal differs significantly from those methods:
>
> (1) The proposed OPNP is training free while Deep Ensemble ans AoP rely on training multiple sparse network from scratch with dynamic sparse regularization [1,2].
>
> (2) We propose a gradient based sensitivity metric for post pruning, whereas those methods utilize the magnitude of features [2] or weights [1,3] as pruning metric. The sensitivitiy based pruning metric is more intuitive and technical sound compared to those magnitude based metric.
>
>  (3) Our proposal prunes both weights and neurons and prunes both the most and the least sensitive units for OOD detection, whereas the existing methods only prunes the weights or features with small magnitude [1,2,3,4].
>
> (4) Our proposal is simple, intuitive, effective, and demonstrate much better performance than DICE [4] (see in table1 and table 2 of our paper).
>
>  (5) We explain why the sensitive based metric is reasonable to improve OOD detection performance, and explains why prunes both the most and the least sensitive units works.
>
> Besides, AoP (released to ArXiv in March) is a contemporaneous work as ours.
>
> Based on our insights and contributions, we believe our work is novel enough and should be known by the community.
> *******************
>
> > **W2:** The pseudocode of Algorithm 1 looks very trivial, and can be significantly improved.
>
> **A:** Thank you for your suggestion. We have improved the pseudocode (see the pdf), and will revise the submission accordingly.
>
> ******************
> We have carefully provided our response to your questions and concerns. We are looking forward to your reply to see whether our response has resolved your concerns. If your have any other questions, please let us know and we are happy to provide more explanations. We will feel grateful if you could boost our paper.
>
> ******************
> **Reference**
>
> [1] Deep ensembling with no overhead for either training or testing: The all-round blessings of dynamic sparsity. ICLR 2022
>
> [2] Extremely simple activation shaping for out-of-distribution detection. ICLR 2022
>
> [3] Average of Pruning: Improving Performance and Stability of Out-of-Distribution Detection. Arxiv 2023.
>
> [4] Dice: Leveraging sparsification for out-of-distribution detection. ECCV 2022.

---

> > ### Comment · Reviewer_Dntp · 2023-08-12
> > **Thanks**
> >
> > I thank the authors for the explanation. I would like to raise the score to Borderline accept.
> >
> > Could the authors explain why pruning weights and neurons that are the most and the least sensitive units for OOD detection is a better choice?
> >
> > Also, could the authors explain what we can observe from the Sensitivity distribution in Figure 2?

---

> > > ### Author Response · Authors · 2023-08-14
> > >
> > > Thank you again for your further review and immediate feedback. And we appreciate that you can increase your rating of this submission. The explanations are provided below.
> > >
> > > > **Q1:**  Could the authors explain why pruning weights and neurons that are the most and the least sensitive units for OOD detection is a better choice?
> > >
> > > **A:** Existing pruning based methods mainly remove the weights or features with low magnitude [1,2,3,4], which is able to avoid overconfident predictions and tends to improve OOD performance. The limitations of those methods are: (1) The magnitude of weights/features is not related to OOD scores directly, which makes magnitude based pruning less intuitive. (2) It is not guaranteed that the separability between ID and OOD samples can be improved by removing smaller weights/features, since the magnitude based metric does not take advantage of prior information about the ID/OOD distributions. (3) Most existing magnitude-based weights pruning are utilized in the training-based methods [2,4], which is relatively high cost.
> > >
> > > In contrast，the proposed sensitivity metric is obtained based on the ID distribution, which is able to identify both the redundant weights (the least sensitive weights) and the risky weights (with exceptionally large sensitivities). The advantagtes of the sensitivity based pruning are: (1) The sensitivity metric is able to make use of prior information about the ID distribution. (2) As the sensitivity metric is measured by the energy score，it better reflects the impact of weights on the OOD scores, making the sensitivity based pruning more intuitive. (3) The separability between ID and OOD distributions can be improved by pruning the least sensitive weights (see **Remark 2**).  (4) The sensitivity based pruning is able to identify and remove the risky weights, which leads to flatter landscape and better generalization (see **Remark 3**). (5) The sensitivity based pruning is low-cost compared to the training-based methods [2,4]
> > >
> > > Therefore the sensitivity based pruning is a better choice to improve OOD detection performance. The experimental results in Table 1 and Table 4 demonstrate the superior of our sensitivity based pruning compared to other pruning methods
> > >
> > > [1] Dice: Leveraging sparsification for out-of-distribution detection. ECCV 2022.
> > >
> > > [2] Deep ensembling with no overhead for either training or testing: The all-round blessings of dynamic sparsity. ICLR 2022.
> > >
> > > [3] Extremely simple activation shaping for out-of-distribution detection. ICLR 2022.
> > >
> > > [4] Average of Pruning: Improving Performance and Stability of Out-of-Distribution Detection. Arxiv 2023.
> > >
> > > ***
> > >
> > > > **Q2:** Could the authors explain what we can observe from the Sensitivity distribution in Figure 2?
> > >
> > > **A:** In our global response (see **Ablations on global pruning**), we demonstrate the performance of pruning the whole model with a global threshold. The sensitivity distribution in Fig. 2 explains why pruning the whole model with a global threshold does not work better than only pruning the last FC layer. Fig.2 shows that the parameter sensitivity of the last FC layer and the shallow Conv layers are not on the same scale (the sensitivity of the FC layer is less than 1/10 of the shallow layers).  This indicates that: (1) Pruning the weights in the Conv layers might lead to model collapse due to extreme high sensitivities. (2) The most sensitive weights in the FC layer can not be pruned when pruning the whole model with a global threshold. (3) There are very few redundant weights (low sensitivity weights) in the Conv layers. Therefore, pruning the weights in the Conv layers does not benefit to OOD detection performance, which explains why pruning the whole model does not work better than only pruning the last FC layer.
> > >
> > > ***
> > >
> > > We are happy to provide further explanations if needed. Thanks!

---

### Author Rebuttal · Authors · 2023-08-09

## Comment
We sincerely thank all the reviewers for their careful reviews and constructive suggestions, which helped us improve our submission. Here, we provide the response to several common concerns and suggestions raised by reviewers.

## Insight Justification
We provide three remarks to explain why OPNP improves OOD detection performance.

**Remark 1: Parameter and neuron pruning avoid overconfident predictions**

**Remark 2: Pruning the least sensitive parameters and neurons improve separability between ID and OOD samples**

**Remark 3: Pruning the most sensitive parameters and neurons improves generalization**

Due to space limitation, we put the details of the remarks in the response to **Review qR61 (R4)**



## Novelty Justification
We declare the contributions and highlights of our work and discuss the relationship between our proposal and other methods

**Contributions and highlights**

(1) We introduce a gradient based approach to estimate the sensitivity of parameters and neurons in deep models and propose the sensitivity based pruning method for OOD detection. The sensitive based pruning is technically sound and we explain why it works

(2) We are the first to prune both the most sensitive and the least sensitive parameters and neurons for OOD detection, and explain the insight behind it.

(3) The whole method is intuitive,  training free, easy to implement and is able to be combined with different post-hoc approaches to achieve promising performance.

(4) Extensive experiments and ablations have been performed to show the effectiveness and efficiency of our proposal.

**The relationship between our proposal and other methods**

We have discussed the relationship between our proposal and Energy Score, ReAct and DICE in Section 3.4. Here, we also discuss the relationship between our proposal and Deep Ensemble [1],  ASH [2] and AoP [3] recommended by Review Dntp. Both Deep Ensemble and AoP rely on training multiple sparse network from scratch with dynamic sparsity constraint,  whereas our OPNP is training free and prunes both the most and the least sensitivity weights and neurons. ASH clips or binarizes the features based on magnitude to improve OOD performance, whereas our OPNP prunes both weights and neurons based on sensitivity. Compared to those methods that also explored sparsity to improve OOD detection performance, our proposal is more intuitive, comprehensive and user-friendly.


## Common Response

**Parameter sensitivity estimation with lower cost**

To reduce the cost for parameter sensitivity estimation, it is advisable to compute the parameter sensitivity based on a subset of the training set rather than the entire set. To demonstrate the effectiveness of using a subset for sensitivity estimation.  We utilize 1%, 5%, 20% and 100% training samples for sensitivity estimation, and perform parameter pruning based on the sensitivities. The experimental results are shown in Table 1, which demonstrate that using only 1% of the training set (ImageNet-1K) also achieved promising performance.

FPR95 and AUROC in SUN and Places datasets are reported, ResNet50 model is utilized. w/o pruning denotes baseline without pruning.
| Sampling Ratio |       w/o pruning | 1%          | 5%          | 20%         | 100%        |
|----------------|-------------------|-------------|-------------|-------------|-------------|
| SUN            | 59.3/85.9         | 32.57/92.83 | 32.06/92.78 | 30.92/93.05 | 30.40/93.17 |
| Places         | 64.9/82.9         | 42.30/90.00 | 41.96/90.08 | 41.21/90.33 | 40.76/90.65 |

**Implementation details of OPNP\+ReAct**

ReAct is a clipping operation that truncates activations above c to limit the effect of noise. It can be defined as $\overline h(x) = \min(h(x), c)$, where $c$ is the clipping threshold. Therefore, ReAct can be integrated into our method easily by clipping the neuron outputs after our neuron pruning stage. The clipping ratio is set to 10% in ResNet50 and 5% in ViT-B/16.

**Ablations on global pruning**

To help understand what happens when weights are pruned globally, we also estimate the parameter sensitivity of the whole model and pruning the weights with different globally pruning stragery. The pruning stargegy are: (1) Global threshold pruning (GTP): pruning all the weights in a global threshold. (2) Layer-wise pruning (LWP): pruning the weights in different layer individually with the same pruning ratio. The results are as follows. We find that both global pruning methods performs worse than pruning the FC layer. We believe the reason is that the sensitivity magnitude across different layers is differs greatly, the sinsitivity of the FC layer is less than 1/10 of the previous Conv layer (See the Fig. 2 in the PDF). Therefore, it is unreasonable to utilize a global threshold for all layers. It might be work if we use different thresholds for different layers, but it is too tricky to determine the optimal thresholds.

| Pruning strategy |    SUN    |   Places  |
|:----------------:|:---------:|:---------:|
|        OPP       | 30.4/93.2 | 40.8/90.7 |
|        GTP       | 39.7/91.9 | 49.1/88.3 |
|        LWP       | 40.5/92.4 | 44.9/90.3 |


**Evaluation Setting**

we utilize the same evaluation setting as ReAct and DICE. The "(subset of) iNaturalist, SUN, Place and Texture" are selected by previous method and widely used for OOD detection evaluation. The hyperparameters is determined in the ID validation set and OOD validation set.

## Reference

[1] Deep ensembling with no overhead for either training or testing: The all-round blessings of dynamic sparsity. ICLR 2022

[2] Extremely simple activation shaping for out-of-distribution detection. ICLR 2022

[3] Average of Pruning: Improving Performance and Stability of Out-of-Distribution Detection. Arxiv 2023.

---

### Decision · Program_Chairs · 2023-09-21

**Decision:**

Accept (poster)

**Comment:**

All reviewers are voting towards a borderline or weak accept, as the work is technically solid and provide improvements upon evaluated methods. Once further results from the rebuttal (e.g. DICE+ReAct) are incorporated into the paper, it will certainly be stronger.